# Top Two Algorithms Revisited

**Marc Jourdan**[1]
marc.jourdan@inria.fr

**Rémy Degenne**[1]
remy.degenne@inria.fr

**Dorian Baudry**[1]
dorian.baudry@inria.fr

**Rianne de Heide**[2]
r.de.heide@vu.nl

**Emilie Kaufmann**[1]
emilie.kaufmann@univ-lille.fr

[1] Univ. Lille, CNRS, Inria, Centrale Lille, UMR 9198-CRIStAL, F-59000 Lille, France
[2] Vrije Universiteit Amsterdam

## Abstract

Top Two algorithms arose as an adaptation of Thompson sampling to best arm identification in multi-armed bandit models [38], for parametric families of arms. They select the next arm to sample from by randomizing among two candidate arms, a *leader* and a *challenger*. Despite their good empirical performance, theoretical guarantees for fixed-confidence best arm identification have only been obtained when the arms are Gaussian with known variances. In this paper, we provide a general analysis of Top Two methods, which identifies desirable properties of the leader, the challenger, and the (possibly non-parametric) distributions of the arms. As a result, we obtain theoretically supported Top Two algorithms for best arm identification with bounded distributions. Our proof method demonstrates in particular that the sampling step used to select the leader inherited from Thompson sampling can be replaced by other choices, like selecting the empirical best arm.

## 1 Introduction

Finding the distribution that has the largest mean by sequentially collecting samples from a pool of candidate distributions ("arms") has been extensively studied in the multi-armed bandit [6, 24] and ranking and selection [21] literature. While existing approaches often rely on parametric assumptions for the distributions, we are interested in (near) optimal and computationally efficient strategies when the distributions belong to an arbitrary class $\mathcal{F}$ of distributions.

For applications to online marketing such as A/B testing [30, 37] assuming Bernoulli or Gaussian arms is fine, but more sophisticated distributions arise in other fields such as agriculture. In Section 5 we consider a crop-management problem: a group of farmers wants to identify the best planting date for a rainfed crop. The reward (crop yield) can be modeled as a complex distribution with multiple modes, but upper bounded by a known yield potential. Therefore, sequentially identifying the best planting date calls for efficient best arm identification algorithms for the class of bounded distributions with a known range.

To tackle this problem, we build on Top Two algorithms [38, 35, 39], originally proposed for specific parametric families. We propose a generic analysis of this type of algorithms, which puts forward new possibilities for the choice of leader and challenger used by the algorithm. In particular, this work leads to the first asymptotically $\beta$-optimal strategies for bounded distributions.

36th Conference on Neural Information Processing Systems (NeurIPS 2022).

## 1.1 Setting and related work

A bandit problem is described by a finite number of probability distributions ($K$ many), called arms. Let $\triangle_K$ be the $K$-dimensional probability simplex and $\mathcal{P}(\mathbb{R})$ the set of probability distributions over $\mathbb{R}$. Let $\mathcal{F} \subset \mathcal{P}(\mathbb{R})$ be a known family of distributions to which the arms to. We will refer to tuples of distributions in $\mathcal{F}^K$ with bold letters, e.g. $\boldsymbol{F} = (F_1, \ldots, F_K) \in \mathcal{F}^K$ where $F_i$ is the cdf of arm $i$. We suppose that all distributions in $\mathcal{F}$ have finite first moment and we denote the mean of $F \in \mathcal{F}$ by $m(F)$. We denote by $\mathcal{I} = \{m(F) \mid F \in \mathcal{F}\}$ the set of possible means for the arms.

The goal of a best arm identification (BAI) algorithm is to identify an arm with highest mean in the set of available arms, i.e. an arm which belongs to the set $i^\star(\boldsymbol{F}) = \arg\max_{k \in [K]} m(F_k)$. At each time $n \in \mathbb{N}$, the algorithm interacts with the environment (the set of arms) by (1) choosing an arm $I_n$ based on previous observations, (2) observing a sample $X_{n,I_n} \sim F_{I_n}$, and (3) deciding whether to stop and return an arm $\hat{i}_n$ or to continue. We study the *fixed confidence* identification setting, in which we require algorithms to make mistakes with probability less than a given $\delta \in (0, 1)$. To compare such algorithms we consider their *sample complexity* $\tau_\delta$, which is a stopping time counting the number of rounds before the algorithm terminates. The goal is then to minimize $\mathbb{E}[\tau_\delta]$ among the class of $\delta$-correct algorithms.

**Definition 1.** *An algorithm is $\delta$-correct[1] on $\mathcal{F}^K$ if $\mathbb{P}_{\boldsymbol{F}}(\tau_\delta < +\infty, \hat{i}_{\tau_\delta} \notin i^\star(\boldsymbol{F})) \leq \delta$ for all $\boldsymbol{F} \in \mathcal{F}^K$.*

In order to be $\delta$-correct on $\mathcal{F}^K$, an algorithm has to be able to distinguish problems in $\mathcal{F}^K$ with different best arms. This intuition is formalized by the lower bound provided in Lemma 1. The characteristic time defined in the lower bound depends on two functions $\mathcal{K}_{\text{inf}}^+$ and $\mathcal{K}_{\text{inf}}^-$, mapping $\mathcal{P}(\mathbb{R}) \times \mathbb{R}$ to $\mathbb{R}_+$, obtained by minimizing a Kullback-Leibler divergence (KL) over $\mathcal{F}$,

$$\mathcal{K}_{\text{inf}}^+(F, u) := \inf\{\text{KL}(F, G) \mid G \in \mathcal{F}, \mathbb{E}_{X \sim G}[X] > u\},$$
$$\mathcal{K}_{\text{inf}}^-(F, u) := \inf\{\text{KL}(F, G) \mid G \in \mathcal{F}, \mathbb{E}_{X \sim G}[X] < u\}.$$

**Lemma 1** (From [16, 3]). *Any algorithm which is $\delta$-correct on $\mathcal{F}^K$ verifies, for any $\boldsymbol{F} \in \mathcal{F}^K$,*

$$\mathbb{E}_{\boldsymbol{F}}[\tau_\delta] \geq T^\star(\boldsymbol{F}) \log(1/(2.4\delta)),$$

*where $T^\star(\boldsymbol{F})^{-1} := \sup_{w \in \triangle_K} \min_{i \neq i^\star} \inf_{u \in \mathcal{I}} \{w_{i^\star} \mathcal{K}_{\text{inf}}^-(F_{i^\star}, u) + w_i \mathcal{K}_{\text{inf}}^+(F_i, u)\}$.*

We say that an algorithm is asymptotically optimal if its sample complexity matches that lower bound, that is if $\limsup_{\delta \to 0} \mathbb{E}_{\boldsymbol{F}}[\tau_\delta]/\log(1/\delta) \leq T^\star(\boldsymbol{F})$.

A related, weaker notion of (asymptotic) optimality is (asymptotic) $\beta$-optimality [39]. An algorithm is called asymptotically $\beta$-optimal if it satisfies $\limsup_{\delta \to 0} \mathbb{E}_{\boldsymbol{F}}[\tau_\delta]/\log(1/\delta) \leq T_\beta^\star(\boldsymbol{F})$, for the complexity term

$$T_\beta^\star(\boldsymbol{F})^{-1} := \sup_{w \in \triangle_K, w_{i^\star} = \beta} \min_{i \neq i^\star} \inf_{u \in \mathcal{I}} \{\beta \mathcal{K}_{\text{inf}}^-(F_{i^\star}, u) + w_i \mathcal{K}_{\text{inf}}^+(F_i, u)\}.$$

An asymptotically $\beta$-optimal algorithm is asymptotically minimizing the sample complexity among algorithms which allocate a $\beta$ fraction of samples to the best arm and $T^\star(\boldsymbol{F}) = \min_{\beta \in (0,1)} T_\beta^\star(\boldsymbol{F})$. As was first shown by [38] when $\mathcal{F}$ is an exponential family, an asymptotically $\beta$-optimal algorithm with $\beta = 1/2$ also has an expected sample complexity which is asymptotically optimal, up to a multiplicative factor 2. That is, $T_{1/2}^\star(\boldsymbol{F}) \leq 2T^\star(\boldsymbol{F})$.

We denote by $w^\star(\boldsymbol{F})$ and $w_\beta^\star(\boldsymbol{F})$ the allocations realizing the argmax in the definition of $T^\star(\boldsymbol{F})$ and $T_\beta^\star(\boldsymbol{F})$, respectively. We will show that for common choices of $\mathcal{F}$ these allocations are unique when there is a unique best arm.

**Distribution classes** The characteristic time $T^\star(\boldsymbol{F})$ depends on the class of distributions $\mathcal{F}$, known to the algorithm in advance, to which $\boldsymbol{F}$ belongs to. For example, all arms could have Bernoulli distributions. We strive to provide an analysis which could easily be applied to many classes $\mathcal{F}$, but we specialize our results to two main cases:

1. distributions with bounded support, $\mathcal{F} = \{F \in \mathcal{P}(\mathbb{R}) \mid \text{supp}(F) \subseteq [0, B]\}$ for $B > 0$,

---

[1] A stronger definition of $\delta$-correctness has also been studied by requiring the algorithm to stop almost surely.

2. single parameter exponential families (SPEF) of sub-exponential distributions.

Given a distribution $\mathbb{P}^{(0)}$ with cumulant generating function $\varphi$, defined on an interval $\mathcal{I}_\varphi$, the SPEF defined by $\mathbb{P}^{(0)}$ is the set of distributions $\mathbb{P}^{(\lambda)}$ with density with respect to $\mathbb{P}^{(0)}$ given by $\frac{d\mathbb{P}^{(\lambda)}}{d\mathbb{P}^{(0)}}(x) = e^{\lambda x - \varphi(\lambda)}$. For example, Gaussian distributions with a known variance form a SPEF, as do Bernoulli distributions with means in $(0,1)$. We consider SPEF of sub-exponential distributions to have a concentration property for the empirical mean estimator.

**Related work**  The first Best Arm Identification (BAI) algorithms [14, 27, 15, 23] were proposed and analyzed for bounded rewards, but their sample complexity scales with a sum of inverse gaps between the means of arms instead of the quantity $T^\star(\boldsymbol{F})$ prescribed by the lower bound. Asymptotically optimal BAI algorithm were first designed when the arms belong to the same single-parameter exponential family. In this context, two families of asymptotically optimal algorithms have emerged. Tracking-based algorithms solve the optimization problem provided by the lower bound in every round, and track the corresponding allocation [16]. The gamification approach views the characteristic time as a min-max game between the learner and the nature, and apply a saddle-point algorithm to solve it sequentially at a lower computational cost [13].

Some Bayesian algorithms arose as another computationally appealing alternative to Track-and-Stop. Russo notably proposed the Top Two Probability Sampling (TTPS) and Top Two Thompson Sampling (TTTS) algorithms [38], that may be seen as counterparts of the popular Thompson Sampling algorithm for regret minimization [41]. Other Bayesian flavored Top Two algorithms have been proposed, Top Two Expected Improvement (TTEI, [35]) and Top Two Transportation Cost (T3C, [39]). All these algorithms sample either a *leader* with fixed probability $\beta$ or a *challenger* with probability $1 - \beta$. TTTS, TTEI and T3C were proved to be asymptotically $\beta$-optimal for Gaussian bandits and perform well in practice even against asymptotically optimal algorithms [35, 39]. This motivates our investigation of Top Two algorithms to tackle bounded distributions, which led us to propose a new generic analysis of this kind of algorithms of independent interest. We prove the asymptotic $\beta$-optimality of several Top Two instances for bounded bandit models, some of which depart from their original Bayesian motivation as they don't need a sampler. An asymptotically optimal algorithm for a non-parametric class of distribution has been proposed by [3] for heavy-tailed rewards. It relies on the computationally prohibitive Track-and-Stop approach, and an adaptation to bounded distributions is mentioned, yet without an explicit calibration of the stopping rule.

## 1.2 Contributions

We present the first fixed-confidence analysis of Top Two algorithms for distribution classes other than Gaussian, including the non-parametric setting of bounded distributions. In Section 2, we introduce several variants of Top Two algorithms, including new ones which choose the empirical best arm as a leader instead of relying on (Thompson) sampling and/or use some penalization in the previously proposed Transportation Cost challenger.

For the class of bounded distributions, we propose in Section 3 a calibration of the stopping rule and a concrete instantiation of the Top Two algorithms, based on a Dirichlet sampler for the randomized variants. We prove in Theorem 1 that those algorithms are asymptotically $\beta$-optimal. This optimality can also be shown for deterministic instances in the case of sub-exponential single parameter exponential families (Appendix H). Our generic analysis, sketched in Section 4, provides insight on what properties the leader and challenger in a Top Two algorithm should have in order to reach asymptotic $\beta$-optimality. We show that the algorithm should ensure that all arms are explored sufficiently, and explain how to guarantee that the sampling proportions reach their optimal values once the sufficient exploration condition holds.

Finally, in Section 5 we report results from numerical experiments on a challenging non-parametric task using real-world data from a crop-management problem for various members of the Top Two family of algorithms. Most of them perform significantly better than the baselines.

## 2 Generic Top Two algorithms

Let $\boldsymbol{F} \in \mathcal{F}^K$ such that $|i^\star(\boldsymbol{F})| = 1$ and $\mu_i := m(F_i) \in \mathcal{I}$ for all $i \in [K]$. As for most BAI algorithms, each arm is pulled once for the initialization. At time $n + 1$, the $\sigma$-algebra $\mathcal{F}_n :=$

$\sigma(U_1, I_1, X_{1,I_1}, \cdots, I_n, X_{n,I_n}, U_{n+1})$, called history, encompasses all the information available to the agent and the internal randomization denoted by $(U_t)_{t \in [n+1]}$, which is independent of everything else. For all $\mathcal{F}_n$-measurable sets $A$, we denote by $\mathbb{P}_{|n}[A] := \mathbb{P}[A \mid \mathcal{F}_n]$ its probability. For an arm $i$, we denote its number of pulls by $N_{n,i} := \sum_{t \in [n]} \mathbb{1}(I_t = i)$, its empirical distribution by $F_{n,i} := \frac{1}{N_{n,i}} \sum_{t \in [n]} \delta_{X_{t,I_t}} \mathbb{1}(I_t = i)$ and its empirical mean by $\mu_{n,i} := m(F_{n,i})$.

**Stopping and recommendation rules**   Our Top Two algorithms rely on the same stopping rule, which can be expressed using the (empirical) transportation cost between arm $i$ and arm $j$, defined as

$$W_n(i,j) = \inf_{x \in \mathcal{I}} \left[ N_{n,i} \mathcal{K}_{\inf}^-(F_{n,i}, x) + N_{n,j} \mathcal{K}_{\inf}^+(F_{n,j}, x) \right]. \tag{1}$$

In particular, using the definition of $\mathcal{K}_{\inf}^\pm$, it can be noted that $W_n(i,j) = 0$ if $\mu_{n,i} \le \mu_{n,j}$. Given a threshold function $c(n, \delta)$, the stopping rule is

$$\tau_\delta = \inf\{n \in \mathbb{N} \mid \min_{j \ne \hat{i}_n} W_n(\hat{i}_n, j) > c(n, \delta)\}, \tag{2}$$

and the recommendation rule is $\hat{i}_n = \arg\max_i \mu_{n,i}$. Up to the choice of threshold, this stopping rule coincides with the GLR-based stopping rule proposed when $\mathcal{F}$ is an exponential family [16] and by [3] for heavy-tailed distributions with an upper bound on a non-centered moment. For a general class $\mathcal{F}$ the stopping rule can be calibrated to ensure $\delta$-correctness under any sampling rule if the threshold is such that the following time-uniform concentration inequality holds for all $\boldsymbol{F} \in \mathcal{F}^K$:

$$\mathbb{P}_{\boldsymbol{F}}\left(\exists n, \exists i \ne i^\star(\boldsymbol{F}): N_{n,i} \mathcal{K}_{\inf}^-(F_{n,i}, \mu_i) + N_{n,i^\star(\boldsymbol{F})} \mathcal{K}_{\inf}^+(F_{n,i^\star(\boldsymbol{F})}, \mu_{i^\star(\boldsymbol{F})}) > c(n, \delta)\right) \le \delta. \tag{3}$$

Lemma 2 in the next section gives an explicit threshold for the class of bounded distribution. For SPEF, we can use generic stopping thresholds derived in [29].

---

1: **Input:** $\beta$
2: Choose a leader $B_n \in [K]$
3: $U \sim \mathcal{U}([0,1])$
4: **if** $U < \beta$ **then**
5:     $I_n = B_n$
6: **else**
7:     Choose a challenger $C_n \in [K] \setminus \{B_n\}$
8:     $I_n = C_n$
9: **end if**
10: **Output**: next arm to sample $I_n$

Figure 1: Generic $\beta$-Top Two sampling rule

Choice of the leader (two propositions):
   **EB** - $B_n^{\text{EB}} \in \arg\max_i \mu_{n-1,i}$
   **TS** - Sample $\theta \sim \Pi_{n-1}$ then set $B_n^{\text{TS}} \in \arg\max_{i \in [K]} \theta_i$

Choice of the challenger (three propositions):
   **TC** - $C_n^{\text{TC}} \in \arg\min_{j \ne B_n} W_{n-1}(B_n, j)$
   **TCI** - $C_n^{\text{TCI}} \in \arg\min_{j \ne B_n} W_{n-1}(B_n, j) + \log N_{n-1,j}$
   **RS** - repeat $\theta \sim \Pi_{n-1}$ until $C_n^{\text{RS}} \in \arg\max_{i \in [K]} \theta_i \not\ni B_n$

Figure 2: Choices of leader and challenger (uniform tie-breaking).

**Sampling rule**   The sampling rule of a Top Two algorithm is shown in Figure 1. The method chooses a first arm $B_n$ called leader which is then sampled with probability $\beta$. If $B_n$ is not sampled, then a second arm $C_n$ called challenger is chosen and sampled. Our analysis isolates properties that those two choices should fulfill in order for the Top Two algorithm to be asymptotically $\beta$-optimal.

The practical implementation of a Top Two method then requires subroutines for $B_n$ and $C_n$. Two possibilities for the leader and three possibilities for the challenger are presented in Figure 2. Our analysis will apply to any combination of those and we will refer to the algorithms obtained by $\beta$-[leader]-[challenger]; for example $\beta$-EB-TCI or $\beta$-TS-TC.

We have two flavors of leaders and challengers: deterministic and randomized. The deterministic choices (EB, for Empirical Best, leader, TC and TCI challengers) rely on the empirical Transportation Costs (TC) $W_n(i,j)$ used in the stopping rule: the TC and TCI challengers are the arms which minimize the transportation cost from the leader (up to a penalization for TCI, hence TC Improved). The randomized choices (TS leader and RS challenger) rely on a *sampler*, denoted by $\Pi_n$. $\Pi_n$ generates i.i.d. vectors $\theta = (\theta_1, \ldots, \theta_K) \in \mathcal{I}^K$ which are interpreted as possible means for the arms, under a distribution which depends on observations gathered in the first $n$ rounds. The TS leader is the best arm in the sampled vector, which is inspired by Thompson Sampling. The RS (for Re-Sampling) challenger is obtained by performing repeated calls to the sampler until the best arm in the sampled vector is not $B_n$, then taking the best arm.

**Randomized instances** The samplers suggested by prior work all have a Bayesian flavor. For SPEF bandits, they use $\Pi_n = \Pi_{n,1} \times \cdots \times \Pi_{n,K}$ where $\Pi_{n,i}$ is the posterior distribution on the mean of arm $i$ after $n$ rounds (given some prior distribution). With this choice of sampler, $\beta$-TS-RS coincides with the TTTS algorithm [38], while $\beta$-TS-TC coincides with the T3C algorithm [39]. TTTS and T3C were only proved to be asymptotically $\beta$-optimal for Gaussian bandits with improper priors, whereas a by-product of the general analysis that we propose in this work permits to establish the necessary properties on the sampler for it to hold for more general distributions. Moreover, we extend these algorithms to bounded distributions by virtue of Dirichlet sampling and also analyze their sampler-free counterparts. As will be apparent in our analysis, the crucial property needed from the sampler in a Top Two algorithm using the RS challenger is that for all arms $i, j$ such that $\mu_i > \mu_j$, $\mathbb{P}_{\theta \sim \Pi_n}(\theta_j > \theta_i) \simeq \exp(-W_n(i, j))$.

**Deterministic instances** Under the RS challenger, the probability to obtain as a challenger arm $j$ is proportional to the probability that $\mathbb{P}_{\theta \sim \Pi_n}(\theta_j > \theta_{B_n})$. Therefore, if $\Pi_n$ is a good sampler satisfying the above property, the TC challenger can be seen as replacing the randomization in the RS challenger by a computation of the mode of the distribution of $C_n^{\text{RS}}$. This was the motivation behind T3C [39] as Gaussian transportation costs have a simple closed form expression while re-sampling becomes more and more costly when the posterior distributions are concentrated. While our asymptotic analysis holds for deterministic algorithms, the empirical performance of fully deterministic algorithms might suffer from unlucky draws. In Section 5, we show that $\beta$-EB-TC is indeed the least robust of all our instances. To cope for this pitfall, explicit or implicit exploration mechanisms can be added. Inspired by IMED [20], the TCI challenger fosters exploration by penalizing over-sampled challengers. Randomization and forced exploration are two other examples of implicit and explicit exploration mechanisms.

## 3 Asymptotically $\beta$-optimal algorithms for bounded distributions

For bounded distribution, Lemma 2 provides a calibration of the stopping rule. Its proof, given in Appendix E.1, relies on a martingale construction proposed by [5].
**Lemma 2.** *The stopping rule (2) with threshold*

$$c(n, \delta) = \log(1/\delta) + 2\log(1 + n/2) + 2 + \log(K - 1) \tag{4}$$

*is $\delta$-correct for the family of bounded distributions.*

**Transportation costs** Both the stopping rule and the TC and TCI challengers of the sampling rule require the computation of $W_n(i, j)$ defined in (1). For single-parameter exponential families, this can be done easily since $\mathcal{K}_{\text{inf}}^{\pm}$ are KL divergences and the transportation cost has a closed form expression [16, 38]. However, for bounded distributions, computing $\mathcal{K}_{\text{inf}}^{\pm}$ is more challenging and we rely on the dual formulation first obtained by [18] (see Theorem 3):

$$N_{n,i}\mathcal{K}_{\text{inf}}^{+}(F_{n,i}, x) = \sup_{\lambda \in [0,1]} \sum_{t \in [n]} \mathbb{1}(I_t = i) \log\left(1 - \lambda \frac{X_{t,i} - x}{B - x}\right) .$$

The minimization in $\lambda$ can be computed using a zero-order optimization algorithm (e.g. Brent's method [10]). The same optimizer can be used to compute the minimization in $x \in [0, B]$ featured in $W_n(i, j)$. By nesting those optimizations of univariate functions on a bounded interval, the computation of $W_n(i, j)$ in the stopping rule dominates the computational cost of our Top Tow algorithms (except the RS challenger). Our experiments suggest that using (2) is twice as computationally expensive as the LUCB-based stopping rule, which is a mild price to pay for the improvement in terms of empirical stopping time. Algorithms for non-parametric distributions are bound to be computationally more expensive than their counterpart in SPEF, where a sufficient statistic can summarize $\mathcal{F}_n$.

**Sampler** The TS leader and RS challenger require a sampler. Our proposed sampler for bounded distributions in $[0, B]$ has a product form: $\Pi_n = \Pi_{n,1} \times \cdots \times \Pi_{n,K}$ where $\Pi_{n,i}$ leverages $\mathcal{H}_{n,i} := (X_{1,i}, \ldots, X_{N_{n,i},i})$, which is the history of samples from arm $i$ collected in the first $n$ rounds. Let $\tilde{F}_{n,i}$ denote the empirical cdf of $\mathcal{H}_{n,i}$ augmented by the known bounds on the support, $\{0, B\}$. For

each arm $i$, $\Pi_{n,i}$ outputs a random re-weighting of $\tilde{F}_{n,i}$. Concretely, letting $\boldsymbol{w} = (w_1, \ldots, w_{N_{n,i}+2})$ be drawn from a Dirichlet distribution $\mathrm{Dir}(\mathbf{1}_{N_{n,i}+2})$, a call to the sampler $\Pi_{n,i}$ returns

$$\sum_{t \in [N_{n,i}]} w_t X_{t,i} + B w_{N_{n,i}+1} .$$

This sampler is inspired by that used in the Non Parametric Thompson Sampling (NPTS) algorithm proposed by [36] for regret minimization in bounded bandits, with the notable difference that we have to add both $0$ and $B$ in the support, while NPTS only adds the upper bound $B$. We will see that this is only necessary to ensure that the re-sampling procedure stops. Therefore, the TS leader could use a sampler $\tilde{\Pi}_n$ based directly on $\mathcal{H}_{n,i}$.

**Theorem 1.** *Combining the stopping rule (2) with threshold (4) and a Top Two algorithm with $\beta \in (0,1)$, instantiated with any pair of leader/challenger as in Figure 2, yields a $\delta$-correct algorithm which is asymptotically $\beta$-optimal for all $\boldsymbol{F} \in \mathcal{F}^K$ with $\mu_{\boldsymbol{F}} \in (0, B)^K$ and $\Delta_{\min}(\boldsymbol{F}) := \min_{i \neq j} |\mu_{F_i} - \mu_{F_j}| > 0$.*

Theorem 1 gives the asymptotic $\beta$-optimality for six algorithms (Figure 2). Choosing our favorite Top Two instances therefore requires further empirical and computational considerations. Computing the EB leader has a constant computational cost, while the TS leader is computationally costly for large time $n$ since it requires to sample from a Dirichlet distribution with $N_{n,i} + 2$ parameters for each arm $i$. On the challenger side, the RS challenger is computationally very expensive for large time $n$ as the sampler becomes concentrated around the true mean vector. On the contrary, by leveraging computations done in the stopping rule (2), the TC and TCI challengers can be computed in constant time. Based on these computational considerations, the most appealing Top Two algorithm for bounded distribution appears to be the fully deterministic $\beta$-EB-TC. But experiments performed in Section 5 reveal its lack of robustness, and for bounded distributions the best trade-off between robustness and computational complexity is $\beta$-EB-TCI. More generally, $\beta$-TS-TC can also be a good choice provided that we have access to an efficient sampler.

**Distinct means** Restricting to instances such that $\Delta_{\min}(\boldsymbol{F}) > 0$ (which implies $|i^\star(\boldsymbol{F})| = 1$) is an uncommon assumption in BAI. However, known Top Two algorithms [38, 35, 39] only have guarantees on those instances. Our generic analysis reveals that it is solely used to prove sufficient exploration, characterized by (7) (Appendix C.3). Experiments highlights that all our Top Two algorithms except $\beta$-EB-TC perform well on instances where $|i^\star(\boldsymbol{F})| = 1$ and $\Delta_{\min}(\boldsymbol{F}) = 0$ (Figure 4(b)). Proving theoretical guarantees in this situation is an interesting problem for future work (see Appendix D.3 for a discussion).

## 4 Sample complexity analysis

In this section, we sketch the proof of Theorem 1, which follows from the generic sample complexity analysis of Top Two algorithms presented in Appendix C. Our proof strategy is the same as that first introduced by [35] for the analysis of TTEI and also used by [39] for TTTS and T3C. It consists in upper bounding the expectation of the *convergence time*, defined as

$$T_\beta^\varepsilon := \inf \left\{ T \geq 1 \mid \forall n \geq T, \ \max_{i \in [K]} \left| \frac{N_{n,i}}{n} - w_i^\beta \right| \leq \varepsilon \right\} , \tag{5}$$

for $\varepsilon$ small enough. Indeed, we prove in Appendix C.5 that for any sampling rule

$$\exists \varepsilon_0(\boldsymbol{F}) > 0, \ \forall \varepsilon \in (0, \varepsilon_0(\boldsymbol{F})], \ \mathbb{E}_{\boldsymbol{F}}[T_\beta^\varepsilon] < +\infty \quad \implies \quad \limsup_{\delta \to 0} \frac{\mathbb{E}_{\boldsymbol{F}}[\tau_\delta]}{\log(1/\delta)} \leq T_\beta^\star(\boldsymbol{F}) . \tag{6}$$

This implication only leverages the expression of the stopping rule and the threshold. It was previously established for Gaussian bandits by [35] and we extend this property to bounded distributions and SPEF of sub-exponential distributions. Up to technicalities ($\mathcal{K}_{\inf}$ continuity and second order terms), this implication is shown by using that if $\tau_\delta \geq n$, then

$$\log(1/\delta) \approx_{\delta \to 0} c(n, \delta) \geq \min_{j \neq \hat{i}_n} W_n(\hat{i}_n, j) \approx_{n \geq T_\beta^\varepsilon} n T_\beta^\star(\boldsymbol{F})^{-1} .$$

To upper bound the expected convergence time, as prior work we first establish *sufficient exploration*:

$$\exists N_1 \text{ s.t. } \mathbb{E}_{\boldsymbol{F}}[N_1] < +\infty, \ \forall n \geq N_1, \quad \min_{i \in [K]} N_{n,i} \geq \sqrt{n/K} . \tag{7}$$

By generalizing [39] which considered Gaussian, we identify two generic properties for the leader and the challenger under which (7) hold (Appendix C.3), provided that we assume $\Delta_{\min} > 0$.

We proceed similarly to prove convergence by identifying in Appendix C desired properties for the leader and challenger, which are satisfied by all our leaders and challengers for bounded distributions (Appendix D). We sketch these conditions below. Let $i^\star$ be the unique element of $i^\star(\boldsymbol{F})$.

The requirements on the leader and the challenger to ensure $\mathbb{E}_{\boldsymbol{F}}[T_\beta^\varepsilon] < +\infty$ become apparent when looking at generic properties of Top Two algorithms. Under any Top Two algorithm, the probability to select arm $i$ at round $n$, $\psi_{n,i} := \mathbb{P}_{|(n-1)}[I_n = i]$, can be written as

$$\psi_{n,i} = \beta \mathbb{P}_{|(n-1)}[B_n = i] + (1 - \beta) \sum_{j \neq i} \mathbb{P}_{|(n-1)}[B_n = j]\mathbb{P}_{|(n-1)}[C_n = i | B_n = j] . \tag{8}$$

We let $\Psi_{n,i} := \sum_{t \in [n]} \psi_{t,i}$. For the leader, we can prove using (8) that

$$\forall M \in \mathbb{N}, \quad \left| \frac{\Psi_{n,i^\star}}{n} - \beta \right| \leq \frac{M-1}{n} + \frac{1}{n} \sum_{t=M}^{n} \mathbb{P}_{|(t-1)}[B_t \neq i^\star] .$$

This suggests that a *good* leader should satisfy that there exists $N_2$ with $\mathbb{E}_{\boldsymbol{F}}[N_2] < +\infty$ s.t.

$$\forall n \geq N_2, \quad \mathbb{P}_{|n}[B_{n+1} \neq i^\star] \leq g(n) , \tag{9}$$

where $g(n) =_{+\infty} o(n^{-\alpha})$ for some $\alpha > 0$. For the challenger, noticing that

$$\forall M \in \mathbb{N}, \forall i \neq i^\star, \quad \frac{\Psi_{n,i}}{n} \leq \frac{M-1}{n} + \frac{1}{n} \sum_{t=M}^{n} \mathbb{P}_{|(t-1)}[B_t \neq i^\star] + \frac{1}{n} \sum_{t=M}^{n} \mathbb{P}_{|(t-1)}[C_t = i | B_t = i^\star],$$

suggests that a *good* challenger should satisfy that there exists $N_3$ with $\mathbb{E}_{\boldsymbol{F}}[N_3] < +\infty$ s.t.

$$\forall n \geq N_3, \forall i \neq i^\star, \quad \frac{\Psi_{n,i}}{n} \geq w_i^\beta + \varepsilon \implies \mathbb{P}_{|n}[C_{n+1} = i | B_{n+1} = i^\star] \leq h(n) , \tag{10}$$

where $h(n) =_{+\infty} o(n^{-\alpha})$ for some $\alpha > 0$. Then, Cesaro's theorem further yields

$$\exists N_4 \text{ s.t. } \mathbb{E}_{\boldsymbol{F}}[N_4] < +\infty, \forall n \geq N_4, \quad \max_{i \in [K]} \left| \frac{\Psi_{n,i}}{n} - w_i^\beta \right| \leq \varepsilon .$$

Using that $(N_{n,i} - \Psi_{n,i})/\sqrt{n}$ are sub-Gaussian random variables, we obtain $\mathbb{E}_{\boldsymbol{F}}[T_\beta^\varepsilon] < +\infty$.

We now explain why (9) and (10) are satisfied for the leaders and challengers in Figure 2 when $\mathcal{F}$ is the class of bounded distributions. This follows from concentration properties. Using the fact that $\sqrt{n}\|F_{n,i} - F\|_\infty$ is sub-Gaussian, which follows for the Dvoretzky–Kiefer–Wolfowitz inequality [31], the continuity of the mean operator $m$ on $\mathcal{F}$ and the sufficient exploration property (7), we establish that for all $\alpha > 0$, there exists a random variable $N_\alpha$ with finite expectation such that

$$\forall n \geq N_\alpha, \quad \max_{i \in [K]} \|F_{n,i} - F_i\|_\infty \leq \alpha \quad \text{and} \quad \max_{i \in [K]} |\mu_{n,i} - \mu_i| \leq \alpha . \tag{11}$$

**Deterministic instances**   Recall that $B_{n+1}^{\text{EB}} \in \arg\max_{i \in [K]} \mu_{n,i}$. Choosing $\alpha$ in (11) smaller than half the gap between the best and second best arm (which is possible as $|i^\star(\boldsymbol{F})| = 1$) yields that for all $n \geq N_\alpha$, $B_{n+1}^{\text{EB}} = i^\star$. This proves (9) with $g(n) = 0$. Using continuity and convexity properties of $\mathcal{K}_{\inf}^{\pm}$, we then establish that there exists $\alpha > 0$ and a problem-dependent constant $C_{\boldsymbol{F}} > 0$ such that for $n \geq N_\alpha$ and for all $i \neq i^\star$,

$$\frac{\Psi_{n,i}}{n} \geq w_i^\beta + \varepsilon \implies \frac{1}{n}\left( W_n(i^\star, i) - \min_{j \neq i^\star} W_n(i^\star, j) \right) \geq C_{\boldsymbol{F}} .$$

This implies that $i \notin \min_{j \neq i^\star} W_n(i^\star, j)$, hence $\mathbb{P}_{|n}[C_{n+1}^{\text{TC}} = i \mid B_{n+1} = i^\star] = 0$ for $n \geq N_\alpha$. Therefore, (10) holds with $h(n) = 0$. A similar argument holds for $C_{n+1}^{\text{TCI}}$.

**Randomized instances**  Let $a_{n+1,i} := \mathbb{P}_{\theta \sim \Pi_n}(i \in \arg\max_{j \in [K]} \theta_j)$ be the probability that arm $i$ is the best arm in a sampled model at round $n$. Since

$$\mathbb{P}_{|n}[B_{n+1}^{\mathrm{TS}} \neq i^\star] \leq (K-1) \max_{i \neq i^\star} a_{n+1,i} \leq (K-1) \max_{i \neq i^\star} \mathbb{P}_{\theta \sim \Pi_n}(\theta_i \geq \theta_{i^\star}),$$

an upper bound on $\mathbb{P}_{\theta \sim \Pi_n}[\theta_i \geq \theta_{i^\star}]$ is sufficient to prove (9). We show in Lemma 64 that this can be obtained by leveraging upper bound on the Boundary Crossing Probability (BCP) of the Dirichlet sampler, $\mathbb{P}_{\theta \sim \Pi_n}[\theta_i \geq u]$ for a fixed threshold $u \in (0, B)$. An upper bound on the BCP can be obtained using the work of [36] and is given in Theorem 5 for the sake of completeness. Putting things together yields that, for all $n$,

$$\mathbb{P}_{\theta \sim \Pi_n}[\theta_i \geq \theta_{i^\star}] \leq f\left(\inf_{u \in [0,B]}[(N_{n,i^\star} + 2)\mathcal{K}_{\mathrm{inf}}^-(\tilde{F}_{n,i^\star}, u) + (N_{n,i} + 2)\mathcal{K}_{\mathrm{inf}}^+(\tilde{F}_{n,i}, u)]\right),$$

where $f(x) = (1+x)e^{-x}$. Using again continuity and concentration (11), we conclude that (9) holds with $g(n) = (K-1)f\left(\left(\sqrt{\frac{n}{K}} + 2\right) D_{\boldsymbol{F}}\right)$, where $D_{\boldsymbol{F}} > 0$ is a problem dependent constant.

For the challenger, we first observe that

$$\mathbb{P}_{|n}[C_{n+1}^{\mathrm{RS}} = i \mid B_{n+1} = i^\star] = \frac{a_{n+1,i}}{1 - a_{n+1,i^\star}} \leq \frac{\mathbb{P}_{\theta \sim \Pi_n}[\theta_i \geq \theta_{i^\star}]}{\max_{j \neq i^\star} \mathbb{P}_{\theta \sim \Pi_n}[\theta_j \geq \theta_{i^\star}]}.$$

Further upper bounding this quantity to prove (10) requires a lower bound on $\mathbb{P}_{\theta \sim \Pi_n}[\theta_i \geq \theta_{i^\star}]$ which can again be obtained using a lower bound on the BCP. In Appendix G.3 we provide a tight lower bound on $\mathbb{P}_{\theta \sim \Pi_n}[\theta_i \geq \theta_{i^\star}]$ featuring the $\mathcal{K}_{\mathrm{inf}}^\pm$ functions. It permits to prove that (10) holds with $-\log(h(n))/n =_{+\infty} \tilde{C}_{\boldsymbol{F}} + o(1)$ where $\tilde{C}_{\boldsymbol{F}} > 0$ is a problem dependent constant.

The above derivations all use the concentration property (11), which requires the sufficient exploration property (7). For our deterministic challengers, sufficient exploration is obtained by noticing that $W_n(i,j)$ can be upper and lower bounded by linear functions of the number of samples. Proving sufficient exploration is more challenging for a randomized challenger, and existing proofs were exploiting the symmetry of the Gaussian posterior. In our analysis we show that a coarse lower bound on the BCP is sufficient to obtain (11), and prove such lower bound for the Dirichlet sampler:

$$\mathbb{P}_{\theta \sim \Pi_n}[\theta_i \geq u] \geq (1 - u/B)^{n+1} \quad \text{and} \quad \mathbb{P}_{\theta \sim \Pi_n}[\theta_i \leq u] \geq (u/B)^{n+1}.$$

These lower bounds ensure that any arm has some (small) probability of being the challenger thanks to re-sampling. Without adding $\{0, B\}$ to $\mathcal{H}_{n,i}$, those probabilities could be equal to zero.

Our analysis is easily amenable to tackle different families of distributions $\mathcal{F}$. This requires continuity and convexity properties for the corresponding $\mathcal{K}_{\mathrm{inf}}$ functions, an appropriate concentration result and further upper and lower bounds on the BCP of the sampler if one wish to analyze randomized algorithms. As an illustration, we show asymptotic $\beta$-optimality of the $\beta$-EB-TC, $\beta$-EB-TCI algorithms for SPEF with sub-exponential distributions, see Appendix H.

## 5  Experiments

We assess the empirical performance of our Top Two algorithms on the DSSAT real-world data and on Bernoulli instances in the moderate regime ($\delta = 0.01$). The stopping rule (2) is used with the threshold $c(n, \delta)$ defined in (4). As Top Two sampling rules, we present results for $\beta$-EB-TC, $\beta$-EB-TCI, $\beta$-TS-TC and $\beta$-TS-TCI with $\beta = 0.5$. Additional experiments are available in Appendix I.2: on the RS challenger whose computational cost prevent it to be evaluated with (4) and on larger sets of arms (up to $K = 1000$).

As benchmarks for the sampling rule, we use KL-LUCB with Bernoulli divergence [28] (whose theoretical guarantees extend to any distribution bounded in $[0, 1]$), "fixed" sampling which is an oracle playing with proportions $w^\star(\boldsymbol{F})$ and uniform sampling. We also propose a heuristic adaptation of the DKM algorithm [13] (which is asymptotically optimal for SPEF) to tackle bounded distributions, which we denote by $\mathcal{K}_{\mathrm{inf}}$-DKM, and uses forced exploration instead of optimism. Inspired by the regret minimization algorithm $\mathcal{K}_{\mathrm{inf}}$-UCB [4], we propose its LUCB variant [27], named $\mathcal{K}_{\mathrm{inf}}$-LUCB. The upper/lower confidence indices are obtained by inverting of $\mathcal{K}_{\mathrm{inf}}^\pm$, i.e.

$$\forall i \neq \hat{i}_n, \quad U_{n+1,i} = \max\left\{u \in [\mu_{n,i}, B] \mid N_{n,i}\mathcal{K}_{\mathrm{inf}}^+(F_{n,i}, u) \leq c(n, \delta)\right\},$$
$$L_{n+1,\hat{i}_n} = \min\left\{u \in [0, \mu_{n,\hat{i}_n}] \mid N_{n,\hat{i}_n}\mathcal{K}_{\mathrm{inf}}^-(F_{n,\hat{i}_n}, u) \leq c(n, \delta)\right\}.$$

LUCB-based algorithms [27] use their own stopping rule, namely they stop when $L_{n+1,\hat{\imath}_n} \geq \max_{j\neq\hat{\imath}_n} U_{n+1,j}$. For Bernoulli distributions, $\mathcal{K}_{\inf}$-LUCB recovers KL-LUCB. While being asymptotically optimal for heavy-tailed distributions [3] with an adequate stopping threshold, the Track-and-Stop algorithm is computationally intractable for bounded distributions as it requires to compute $w^\star(\boldsymbol{F}_n)$ at each time $n$ (or on a geometric grid). We hence omit it from our experiments.

**Crop-management problem** We benchmark our algorithms on the DSSAT simulator[2] [22]. Each arm corresponds to a choice of planting date and fixed soil conditions (details in Appendix I). To illustrate the problem's difficulty we represent an empirical estimate (independent of the runs of our algorithms) of the yield distributions in Figure 3(b). Since the gaps between means are small, the identification problem is hard. Moreover, $\mathcal{K}_{\inf}$ computations for non-parametric distributions are costlier than Bernoulli ones (see Appendix I.1), so we only present the results for 100 runs.

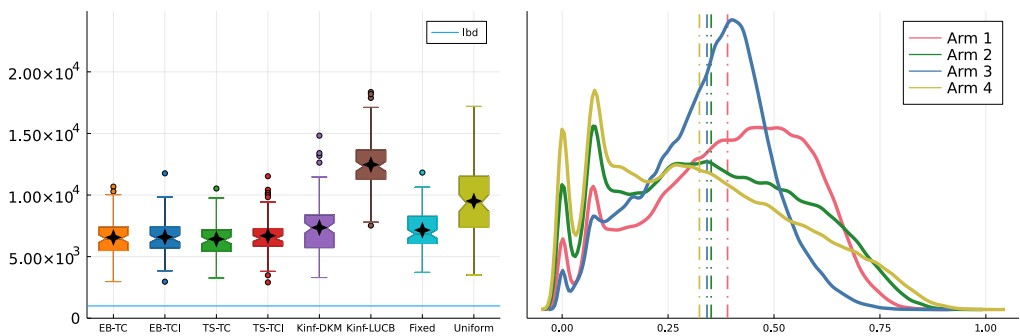

Figure 3: Empirical stopping time (a) on scaled DSSAT instances with their density and mean (b). Lower bound is $T^\star(\boldsymbol{F})\log(1/\delta)$. "stars" equal means.

In Figure 3, $\beta$-EB-TCI, $\beta$-TS-TC and $\beta$-TS-TCI slightly outperform $\mathcal{K}_{\inf}$-DKM and the fixed (oracle) sampling rule. Moreover, $\mathcal{K}_{\inf}$-LUCB performs significantly worse than uniform sampling. Due to the small number of runs, we don't observe large outliers for $\beta$-EB-TC (see Appendix I.2). KL-LUCB performs ten times worse than $\mathcal{K}_{\inf}$-LUCB, hence we omit it from Figure 3.

**Bernoulli instances** Next we assess the performance on 1000 random Bernoulli instances with $K = 10$ such that $\mu_1 = 0.6$ and $\mu_i \sim \mathcal{U}([0.2, 0.5])$ for all $i \neq 1$, where we enforce that $\Delta_{\min} \geq 0.01$. We also study the instance $\mu = (0.5, 0.45, 0.45)$, in which $\Delta_{\min} = 0$, and perform 1000 runs.

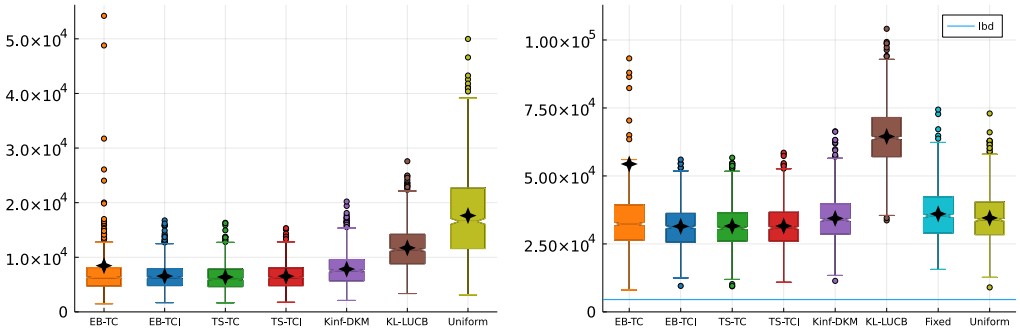

Figure 4: Empirical stopping time on Bernoulli (a) random instances with $K = 10$ and (b) instance $\mu = (0.5, 0.45, 0.45)$.

In Figure 4(a), we see that $\beta$-EB-TCI, $\beta$-TS-TC and $\beta$-TS-TCI outperform other algorithms. While this gain is slim compared to $\mathcal{K}_{\inf}$-DKM, the empirical stopping time is twice (resp. three times) as large for KL-LUCB (resp. uniform sampling). Even when $\Delta_{\min} = 0$, Figure 4(b) hints that their empirical performance might be preserved. Figure 4 confirms the lack of robustness of $\beta$-EB-TC,

[2]DSSAT is an Open-Source project maintained by the DSSAT Foundation, see https://dssat.net.

which is prone to large outliers. For the symmetric instance in Figure 4(b), uniform sampling outperforms KL-LUCB and perform on par with the "fixed" sampling.

## 6   Perspectives

We provided a general analysis of Top Two algorithms, including new variants using the EB leader and TCI challenger, and proved their asymptotic $\beta$-optimality on the non-parametric class of bounded distributions. On experiments on distributions coming from a real world application, several Top Two variants (in particular $\beta$-TS-TC and $\beta$-EB-TCI) proved more effective than all baselines. Furthermore, $\beta$-EB-TCI is computationally not costlier than computing the stopping rule.

As in previous work on Top Two methods our result only characterizes the asymptotic performance of the algorithms, and obtaining bounds on the sample complexity for any $\delta$ that would reflect their good empirical performance is a most pressing open question. Our work also hints at what is needed to obtain non-asymptotic guarantees: the only variant for which the empirical behavior does not reflect the asymptotic bound is $\beta$-EB-TC, which is also the most greedy variant. Algorithms using a sampler naturally explore, and the penalized version $\beta$-EB-TCI successfully corrects the shortcomings of $\beta$-EB-TC by penalizing over-sampling. Quantifying the amount of exploration required by Top Two algorithms should also allow the removal of the hypothesis $\Delta_{\min} > 0$ from Theorem 1.

Finally, Top Two algorithms are promising algorithms to tackle the setting of fixed budget identification, in which the algorithms have to stop at a given time and should then make as few mistakes as possible. As their sampling rule is anytime (i.e. independent of $\delta$), Top Two algorithms might also have theoretical guarantees for BAI in the fixed-budget setting or even the anytime one, in which guarantees on the error probability should be given at all time.

## Acknowledgments and Disclosure of Funding

Experiments presented in this paper were carried out using the Grid'5000 testbed, supported by a scientific interest group hosted by Inria and including CNRS, RENATER and several Universities as well as other organizations (see https://www.grid5000.fr). This work has been partially supported by the THIA ANR program "AI_PhD@Lille". The authors acknowledge the funding of the French National Research Agency under the project BOLD (ANR-19-CE23-0026-04), and the Dutch Research Council (NWO) Rubicon grant number 019.202EN.004.

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
