# OpenReview forum: "Top Two Algorithms Revisited"
_NeurIPS.cc/2022/Conference — NeurIPS 2022 Accept_

### Official Review · Reviewer_166Y · 2022-07-11

**Rating:** 6
**Confidence:** 3
**Soundness:** 2 fair
**Presentation:** 2 fair
**Contribution:** 3 good

**Summary:**

The authors improve the top-two sampling algorithm, commonly used as bayesian BAI strategies. The authors present asymptotic optimality of the BAI strategy for nonparametric bandits with bounded support, which previous work has not shown.

**Questions:**

None.

**Limitations:**

Although the contribution is significant, I feel that the results presented are not so surprising, given the context of existing research.

**Strengths And Weaknesses:**

This study generalizes Top Two strategies, a well-known Bayesian BAI strategies. This contribution is very important. In recent years, the Top Two strategy has attracted much attention due to its ease of use, and the authors' results reinforce its advantages. In particular, the results for bounded nonparametric distributions are interesting and also contribute to a whole bandit study.

---

> ### Author Response · Authors · 2022-08-01
> **Answer to Reviewer 166Y**
>
>
> We thank the reviewer for their time and comments. We do believe that there are some surprising elements in our work:
>
> - Existing Top Two algorithms were all motivated by a Bayesian philosophy, mostly fixing Thompson Sampling to make it work for BAI. Many of them rely on randomization (posterior sampling). We propose new deterministic choices for both the leader and challenger. In particular, we advertise the use of EB-TCI which performs exploration in the challenger by replacing (costly) randomization by a deterministic penalization.
>
> - Our proposed Top Two algorithms for bounded distributions are not straightforward. First, they require to introduce an appropriate sampler for stochastic variants (a Dirichlet sampler similar to those used in recent algorithms for regret but with the twist of adding both 0 and B in the history). Second, to justify the use of the TC challenger as an approximation to RS (and analyze it), we had to prove tight bounds on the probability that on Dirichlet sample exceeds another, which are highly non trivial (see Appendix G).
>
> Furthermore we emphasize that one of our key contributions is to provide a unified analysis of Top Two algorithms, which decouples the properties needed from the leader and the challenger and further identifies what assumptions are needed on the rewards distributions. All these elements contribute to a better understanding of the increasingly popular Top Two methods.

---

> > ### Comment · Reviewer_166Y · 2022-08-08
> > **Response to the authors' reply**
> >
> > Thanks for your detailed comments. I have read your reply. Although I am still wondering whether this study introduces a significantly novel idea to the BAI literature, I admit the novelty and contribution of this study in that it updates our knowledge of the Top two sampling algorithms.
> >
> > Let me ask you the following additional questions:
> > 1. The prior work of the Top two sampling methods are Bayesian, but the methods of this work are not restricted to Bayesian but includes frequentist flavored methods. This is one of the contributions of this work. Is my understanding correct?
> > 2. Why is there a difficulty in $\beta$-optimality only for the bandits with bounded supports? Shang et al. (2019) did not show the $\beta$-optimality only for Gaussian bandits, while they showed the optimalities for both Gaussian and Bernoulli bandits. I thought it is straightforward obtaining the $\beta$-optimality at least for Bernoulli bandits, based on Shang et al. (2019) 's results. Could you give me an intuition?
> > 3. As other reviewers may have pointed out, I think the notations of the $beta$ indicating the assignment of samples to the best arm and the $beta$ for the threshold are a bit confusing.
> > 4. Are there any restrictions on the prior choice? To the best of my knowledge, in Russo's original result, for example, we cannot use Beta(1, 1) as the prior. Later, Shang et al. (2019) relaxed the restriction. It seems that this study uses a Dirichlet prior to the TS leader. Can we use a wide class of the prior distributions?
> > 5. The authors show a $\beta$ optimality, which is commonly used in Bayesian BAI literature. Could you show global optimality, such as Garivier and Kaufmann (2016)? If not, could you explain the reason?
> >
> > Also, as a minor comment, I think "Finally, Top Two algorithms have not been studied in the setting of fixed budget identification, in which the algorithms have to stop at a given time and should then make as few mistakes as possible." in Section 6 is a bit strongly worded. For example, in econometrics, Kasy and Sautmann (2021a) adapted Top Two Thompson sampling to a fixed-budget setting. Although there is a technical error in the arguments of the asymptotic optimality, Ariu et al. (2021) and Kasy and Sautmann (2021b) correct the error based on the anytime optimality.  I think that you do not need to cite these studies. However, since prior work exists and other work still may exist, it would be nice to weaken the wording.
> >
> > Kasy and Sautmann (2021a), "Adaptive Treatment Assignment in Experiments for Policy Choice," Econometrica
> > Kasy and Sautmann (2021b), "Correction regarding "Adaptive treatment assignment in experiments for policy choice."
> > Ariu, Kato, Komiyama, McAlinn, and Qin (2021), "Policy choice and best arm identification: Asymptotic analysis of exploration sampling."

---

> > > ### Author Response · Authors · 2022-08-08
> > > **Response to Reviewer 166Y's reply**
> > >
> > > Thank you for your comments and questions. Please find answers to your 5 questions below.
> > >
> > > 1. Among the proposed Top Two algorithms, one of our contributions is indeed to propose a frequentist choice for the leader. The EB leader is the first deterministic leader proposed so far.
> > > In Russo (2016), the leader and challenger were Bayesian. While the challenger was frequentist in Shang et al. (2019), their leader was still Bayesian.
> > >
> > > 2. In Shang et al. (2019), they show that the posterior converges with the $\beta$-optimal rate for Bernoulli and Gaussian bandits (Theorem 4-5).
> > > We do not explore this line of guarantees and focus on the expected sample complexity.
> > > While they prove asymptotic $\beta$-optimality for Gaussian bandits (Theorem 1), they do not provide the same guaranty for Bernoulli bandits.
> > > For Bernoulli bandits, the Dirichlet sampler we propose coincide with their Beta posterior, hence we prove a sample complexity bound on their algorithm.
> > > Their proof of $\beta$-optimality for Gaussian bandits relies heavily on specific properties of Gaussians: concentration, explicit formulas, symmetry of the posterior, approximation of $\mathbb P_{\theta \sim \Pi_{n}}(\theta_j > \theta_{i})$, etc.
> > > Therefore, even for Bernoulli, it is not straightforward to adapt them. Although since Bernoulli distributions are still a simple parametric family, a proof along the same lines could perhaps give a bound.
> > > However, the class of bounded distributions is non-parametric and some properties used for the parametric case of Gaussians don't apply.
> > > Our unified analysis is agnostic to the considered distributions and highlights the different technical difficulties.
> > > This allowed us to conclude the proof even for the non-parametric case of bounded distributions.
> > >
> > > 3. We acknowledge that the two $\beta$ notations can be confusing. They were inherited from prior work, and we will consider changing one of them.
> > >
> > > 4. Our approach is not Bayesian in nature and the samplers we propose (for the Top Two variants that use sampling) don't necessarily correspond to a posterior, hence don't correspond to a choice of prior.
> > > Our algorithm $\beta$-EB-TCI has no Bayesian component: since it is frequentist in the choices of the leader and challenger, there is no sampler or prior.
> > > For Bernoulli distributions, the Dirichlet sampler we use in TS and RS coincides with the Beta posterior using as prior the uniform distribution over {$0,B$}.
> > > For non-parametric distributions, it is not clear whether the Dirichlet sampler can be seen as a posterior.
> > > Our approach was not to analyze a given prior/posterior to use in posterior sampling, but to identify the properties we needed in a sampler and then come up with a suitable one.
> > > In particular, we show that for bounded distributions the Dirichlet sampler satisfies $\mathbb P_{\theta \sim \Pi_{n}}(\theta_j > \theta_{i}) \approx \exp(- W_n(i,j))$, and we prove that this property allows to obtain asymptotic $\beta$-optimality of our proposed sampling-based Top Two algorithms.
> > >
> > > 5. Based on the theoretical lower bound, Top Two algorithms with a fixed allocation $\beta$ can be at best $\beta$-optimal, not globally optimal.
> > > To achieve global optimality, the fixed allocation should match the optimal allocation $\beta^\star = argmin_{\beta \in (0,1)} T^\star_{\beta}( F)$.
> > > As $\beta^\star$ is unknown, it should be learned from observations.
> > > Therefore, this desired adaptive Top Two algorithm should use an adaptive choice of $\beta$ which converges towards $\beta^\star$.
> > > Proving optimality for adaptive Top Two algorithms is an interesting open problem, which is still unsolved even for Gaussian bandits. The very recent paper [Optimality Conditions and Algorithms for Top-K Arm Identification, Wang et al, 2022] proposes an update mechanism for $\beta$, but they study it only empirically and they don't provide any theoretical guaranty for that scheme.
> > >
> > > Finally, we thank you for the new references on Top Two algorithms for fixed budget.
> > > We will cite those papers and change the way we refer to the fixed budget setting to reflect those works.

---

> > > > ### Comment · Reviewer_166Y · 2022-08-09
> > > > **Thank you for you clarifiation**
> > > >
> > > > Thanks for your quick reply and clarification! The reply from the authors answered my question well. Although I still have some concerns about the impact of the findings, I agree with the novelty and contribution. Now I lean toward acceptance and will re-evaluate my rating after checking the paper again.

---

### Official Review · Reviewer_dAwE · 2022-07-17

**Rating:** 7
**Confidence:** 3
**Soundness:** 3 good
**Presentation:** 3 good
**Contribution:** 3 good

**Summary:**

This paper investigates Top Two algorithms, which are a series of algorithms aiming to find the best arm in the multi-armed bandit problems with fixed confidence. The main contribution of this work is two-fold: 1) It demonstrates the first Top Two method with theoretical guarantee ($\beta$-optimality) when the input arms have bounded rewards; 2) It also offers a generic analysis that defines desirable characteristics a Top Two algorithm need to satisfy in order to have theoretical guarantee.

**Questions:**

With the exception of the following errors and concerns, I believe this work to be solid and one that could expand the possibilities for Top Two algorithms.

Line 32: "... which the arms belong.": belong -> belong to

Line 44: The definition of $\delta$-correctness seems to me that it misses the scenario when an algorithm could run in infinite time.

Line 47: $\mathcal{P}(\mathbb{R})$ is not defined in the previous sections

Line 50: $\Delta_k$ is not defined in the previous sections, which makes me hard to understand the definition of $T^{*}(\textbf{F})$. Besides, it would be preferable if the author(s) could briefly explain Lemma 1 afterward because it is essential to comprehending the work.

Line 56: Should it be $T^*(\mathbf{F}) = \min_{\beta \in (0,1)} T_{\beta}^*(\mathbf{F})$ ?

Line 64: "... to which $\mathbf{F}$ belongs.": belongs -> belongs to

**Limitations:**

N/A since there isn't any negative societal impact.

**Strengths And Weaknesses:**

Strengths

* This is the first Top Two algorithm in bandit fields which has theoretical guarantee on the problems with input arms having bounded rewards, which is novel.
* The analysis approach can be extended to more general reward distributions as long as the algorithm has the properties proposed in the paper.
* Experiment results using real-world dataset are presented to demonstrate the superior empirical performance of the proposed algorithms against the baselines.

Weaknesses

* The proof can only be applied when the input instance has only one best arm. It is an interesting direction to see if this assumption could be removed.
* I think the presentation could be a little bit better, for example, some definitions or lemmas are lacking explanations before they are introduced.

---

> ### Author Response · Authors · 2022-08-01
> **Answer to Reviewer dAwE**
>
> We thank the reviewer for their comments and suggestions to improve the presentation, which we will incorporate in our revision. We provide below an answer to two specific concerns.
>
> **Unique best-arm assumption**
> To the best of our knowledge, all the literature on (exact) fixed-confidence BAI assumes that the instance has only one best arm. Indeed, in the presence of two best arms existing algorithms would only stop with a small probability as they would try to statistically distinguish two identical distributions. However, a natural question following your remark is whether (an adaptation) our top two algorithms could be used for a relaxation of the BAI problem in which the goal is to find one arm which is $\varepsilon$-close to the best arm, for some parameter $\varepsilon>0$. When it comes to asymptotic optimality, this setting is known to be much more complex than standard BAI, see e.g. (Degenne and Koolen, 2019, Pure Exploration with Multiple Correct Answers). Using Top Two algorithms in this setting would require a different stopping rule but also an appropriate definition of the leader to explore all the possible candidate best arms. We leave this extension as an interesting direction for future work.
>
> **Definition of $\delta$-correctness** Two alternative definitions of $\delta$ correctness have been used in the literature. The one we chose (also used in other works) allows to obtain $\delta$-correctness solely based on the stopping and recommendation rules, independently of the sampling rule.
> In order to prevent the algorithm to run infinitely (and give guarantees on the sample complexity), the sampling rule has of course to be factored in.
> This body of literature is interested in upper bounding the expectation of the stopping time, and a finite expectation implies in particular that the algorithm stops almost surely.
> The alternative definition of $\delta$-correctness consists in assuming that $\mathbb P_{F}(\tau_{\delta} < +\infty) = 1$ and $\mathbb P_{F}(\hat \imath_{\tau_{\delta}} \notin i^\star(F)) \le \delta$. Both definitions are used in the literature, but we prefer the former one for the above reason.

---

### Official Review · Reviewer_KZTG · 2022-07-18

**Rating:** 4
**Confidence:** 4
**Soundness:** 3 good
**Presentation:** 3 good
**Contribution:** 2 fair

**Summary:**

This paper analyzes several top-two algorithms for more general distributions, including those from single parameter exponential families and non-parametric bounded distributions.

**Questions:**

1. What does I in TCI refer to? What is the difference between challengers TCI and TC(I)?

2. It seems that algorithms using RS as the challenger is not included in the experiments. Is the reason that running RS could be time-consuming? I guess due to the randomness, algorithms using RS might perform even better than others included in the experiments. For Gaussian, we have good approximations of RS. Is it possible to come up with approximations of RS for general distributions studied in the paper?


**Limitations:**

The conclusion section describes some open problems and future directions. I think the contributions of the paper can be improved if the analysis of some top-two algorithms could be generalized to cover the case Delta_min = 0.

**Strengths And Weaknesses:**

In terms of theoretical contributions, the analysis in this paper extends the previous analysis for Gaussian distributions. Although the paper introduces several algorithms, I am not sure they are novel. In my opinion, it is well-known that other leaders, e.g., the empirical best arm also work for top-two algorithms as well as the theoretical analysis. Indeed, when the leader is the empirical best arm, the analysis becomes easier than that for the leader chosen by Thompson sampling (TS). In terms of empirical results, it is also known that the randomness in TS or resampling helps exploration, while top-two algorithms directly using the posterior quantities might be less efficient since they can be inaccurate due to the random and unlucky observations.

---

> ### Author Response · Authors · 2022-08-01
> **Answer to Reviewer KZTG**
>
>
> We thank the reviewer for their time and comments.
>
> **Algorithmic novelties** Our work contain several algorithmic novelties.
>
> - We respectfully disagree with the statement that ``it is well-known that other leaders, e.g., the empirical best arm also work for top-two algorithms as well as the theoretical analysis''. To the best of our knowledge only two leaders have been analyzed in the fixed confidence setting: expected improvement in TTEI (which is specific to Gaussian) and the TS leader (proposed for any SPEF and analyzed in the Gaussian case). Top-Two algorithms were originally introduced as an adaptation of Thompson Sampling to best arm identification, and the TS leader could easily be perceived as central to their performance. Our analysis reveals that it is not.
>
> - While inspired by the TC challenger (Shang et al 2019) and the IMED algorithm (Honda and Takemura 2015), the TCI challenger is novel. The good empirical performance of EB-TCI (see Figures 5-8) reveals that the (costly) randomization fostering exploration can be replaced by an appropriate penalization.
>
> - Our instantiations of Top Two algorithms for bounded distributions are novel. They require to define an appropriate sampler in a non-parametric setting, and justify the TC approximation (see below). Introduced by Riou and Honda (2020) for regret minimization, the Dirichlet sampler is very recent in the bandit literature.
> We proved it could be successfully adapted to BAI.
>
>
> **Technical novelties** Our analysis is not a mere extension from the Gaussian case. We propose a unified and modular analysis which explicitly sheds light on the properties that the leader and the challenger should satisfy to obtain sufficient exploration and convergence towards the optimal allocation.
> This allows us to prove that other leaders and challengers are possible, including the EB leader.
> In the previous analyses of Gaussian top-two algorithms, the different steps of the proof were highly intertwined, hence supporting deterministic choices of leader was not straightforward.
> Our analysis was used to prove $\beta$-optimality for non-parametric bounded distributions, which is a significant contribution. Besides the proof structure, we had to change many arguments that were specific to Gaussian (symmetry of the posterior, lower bounds on tail probabilities).
>
>
> **``What does I in TCI refer to?''**
> It refers to ``Improved'' (line 1010) and it was meant to be reminiscent of the I in IMED which uses a similar penalization in a different context.
> TC(I) refers simultaneously to TC and TCI, and that will be clarified in the final version.
>
> **Experiments with the RS challenger**
> Due to its computational cost, the RS challenger was not included in the experiments reported in the main content.
> In Appendix I.2, we perform experiments with $\beta$-EB-RS and $\beta$-TS-RS (Figures 5, 6, 7, 8 and Table 3).
> Those experiments could terminate because we used a smaller heuristic threshold instead of the theoretically valid threshold defined in (4) which is used in Section 5.
> Based on these experiments, the RS and TCI challengers perform on par.
>
> The computational cost of RS explodes for large time $n$ because the sampler becomes concentrated around the true mean vector and as a consequence the number of re-sampling steps before finding a different leader can be very large.
> Even on Bernoulli instances (Figures 5, 6) for which an efficient sampler exists, RS requires more than $10^6$ re-sampling steps at each late iteration, hence yielding a $10^4$ higher computational time.
>
> **Approximation of RS for general distributions** You suggest that we could approximate the RS challenger and we agree: this is precisely the idea behind TC/TCI.
> In Appendix G, we show that the Dirichlet sampler verifies $\mathbb P_{\theta \sim \Pi_{n}}(\theta_j > \theta_{i}) \approx \exp(- W_n(i,j))$ for bounded distributions. Using this approximation $C_n^{\text{TC}}$ coincides with the mode of the distribution of $C_n^{\text{RS}}$. See also the explanation in the last paragraph of Section 2.
> For the Gaussian sampler, the upper and lower bound to obtain this approximation are both easy to obtain.
> However, for the Dirichlet sampler, deriving this approximation was challenging and is a contribution of independent interest.
>
> **Instances with $\Delta_{\min} (F)=0$**
> The case $\Delta_{\min} (F)=0$ is discussed in Appendix D.3.
> Thanks to our analysis, we show that the assumption $\Delta_{\min} (F) > 0$ (used by all previous work) is only used to show sufficient exploration.
> Experiments (Figure 4(b) and 8) suggest that this assumption could (and should) be removed for Top-Two algorithms having a randomization or a penalization mechanism.
> Formally showing this intuition is an interesting open problem.
> We note that this assumption is unnecessary if we add forced exploration, but this is empirically wasteful and we would prefer to avoid it.

---

### Official Review · Reviewer_yFSX · 2022-07-18

**Rating:** 7
**Confidence:** 4
**Soundness:** 3 good
**Presentation:** 4 excellent
**Contribution:** 3 good

**Summary:**

This paper provides an in-depth analysis of “Top-Two” style algorithms for the pure-exploration multi-armed bandit problem under bounded distributions. The authors consider a variety of methods to choose the leader and challenger which recover many of the well known algorithms already in the literature - such as TTTS and T3C. They provide a general theory for analyzing the sample complexity of top-two algorithms and also provide some experiments.


**Questions:**

I was a bit surprised that KL-LUCB was worse than uniform in Figure 4b. Can you shine some light on this?


**Limitations:**

None.

**Strengths And Weaknesses:**

Strengths:
For bounded distributions, this paper provides a general purpose methodology to derive asymptotically-optimal algorithms which do not require computationally expensive track and stop style forced exploration. From that perspective, I think this work is useful for practitioners - indeed most practical use cases are bounded distributions, and sub-Gaussian approximations can be woefully sub-optimal.

I thought the paper was well written and thorough.

Weaknesses:
Since this paper is providing such a thorough investigation of Top-Two style algorithms, I would have liked to see some more experiments with a larger set of arms. For example on some of the instances that have been benchmarked in previous literature, i.e. Jamieson, Kevin, and Robert Nowak. "Best-arm identification algorithms for multi-armed bandits in the fixed confidence setting." 2014 48th Annual Conference on Information Sciences and Systems (CISS). IEEE, 2014.

My second comment is about the lack of finite-time non-asymptotic guarantees. I think at this point, there is plenty of literature in the asymptotic setting. The authors address this limitation so I do not really hold this against them.

Finally - just to really point out novelty, I think the authors can better indicate what aspects of their analysis are novel in comparison to past works.

---

> ### Author Response · Authors · 2022-08-01
> **Answer to Reviewer yFSX**
>
>
> We thank the reviewer for their time and comments.
>
> **Experiments on large set of arms**
> We would like to thank the reviewer for this great suggestion.
> We uploaded a revised version of the supplementary material, which contains new (preliminary) experiments in Appendix I.2.3 (Figure 9).
> As advised, we consider the three Gaussian benchmarks of Jamieson and Nowak (2014) for varying number of arms (up to $1000$).
> Thanks to this new insightful experiment, we see that $\beta$-TS-TC has the best scaling with the number of arms and outperforms other methods.
> Moreover, we observe that the performance gap between $\beta$-EB-TCI and LUCB is decreasing, and for larger sets of arms LUCB slightly outperforms it.
> More detailed experiments will be added in the final version, in particular including other benchmarks.
>
> In Appendix I.2, additional experiments were performed for the real-world DSSAT data (up to 6 arms) and for random Bernoulli instances (up to 10 arms).
>
> **Empirical performance of KL-LUCB vs uniform sampling**
> In Figure 4(b), the considered Bernoulli instance is $\mu = (0.5, 0.45, 0.45)$, hence the two sub-optimal arms have the same mean.
> By symmetry of the characteristic time $T^\star(\mu)$, we have $w^\star(\mu)_2 = w^\star(\mu)_3 = (1 - w^\star(\mu)_1) / 2$, and in fact that optimal allocation is close to uniform for that instance.
> Experimental results (Figure 4(b) and Figure 8) show that the uniform sampling performs on par with the fixed oracle algorithm tracking $w^\star(\mu)$.
> Therefore, it is not surprising that KL-LUCB performs worse.
> On random and non-symmetric Bernoulli instances (Figure 4(a), 5 and 6), KL-LUCB doesn't perform worse than uniform sampling.
>
> **Finite-time**
> The lack of finite-time non-asymptotic guarantees for Top-Two algorithms is indeed a most pressing open question on which we are currently working.
>
> **Novel technical results**
> In the revised version, we will better indicate the aspects of the analysis that are novel. For reference, here is a short list:
>
> - The general proof strategy is inherited from Qin et al (2017) and Shang et al (2019), but its different steps were highly intertwined and a change of leader for example could need changes at many places. We transformed that proof technique into a unified and modular analysis, which translates into a reduced proof burden for each variant.
>
> - Our analysis applies to non-parametric bounded distributions, as you noted in the review. It also applies to many single-parameter exponential families, while only Gaussians were covered in previous work.
>
> - This required to change many arguments that were specific to Gaussian. As an example, our proof of Lemma 25 holds under very generic assumptions on the boundary crossing probability for the sampler (Property 9 and 10) whereas the counterpart of this proof in the Gaussian case exploits both the symmetry of the Gaussian posterior and well known tail bounds for Gaussian distributions.
>
> - To prove $\beta$-optimality for bounded distributions, we proved novel deviation bounds on a Dirichlet sampler (see Appendix G) as well as new properties of the Kinf function (Appendix F).

---

> > ### Comment · Reviewer_yFSX · 2022-08-09
> > **Thank you for the response**
> >
> > Thank you for the response. I am satisfied with your remarks and additional experiments. I'm revising my score to Accept.

---

### Meta-Review · Area_Chair_ALuw · 2022-08-20

**Recommendation:** Accept
**Confidence:** Less certain

**Metareview:**

This paper analyzes several "top-two" style algorithms for the pure-exploration multi-armed bandit problem under bounded distributions.  The reviewers agreed that the theoretical contribution of this paper is solid. Some concern was raised on the algorithmic and empirical contributions of this paper.  In particular, it was mentioned by one reviewer that deterministic choices of leader and challenger might also work, and the analysis might be easier.  I hope this item can be addressed before this paper is published.

**Award:**

No

---

### Decision · Program_Chairs · 2022-09-14

Accept